# Evaluation of the Hematological Patterns from Up to 985 Days of Long COVID: A Cross-Sectional Study

**DOI:** 10.3390/v15040879

**Published:** 2023-03-29

**Authors:** Vanessa Costa Alves Galúcio, Daniel Carvalho de Menezes, Patrícia Danielle Lima de Lima, Vera Regina da Cunha Menezes Palácios, Pedro Fernando da Costa Vasconcelos, Juarez Antônio Simões Quaresma, Luiz Fábio Magno Falcão

**Affiliations:** 1Center for Biological Health Sciences, State University of Pará (UEPA), Belém 66087-670, Brazil; 2School of Medicine, São Paulo University (USP), São Paulo 01246903, Brazil; 3Tropical Medicine Center, Federal University of Pará (UFPA), Belém 66055-240, Brazil

**Keywords:** long COVID, laboratory markers, hematological tests, blood cell count, erythrocyte indices

## Abstract

Long COVID affects many individuals following acute coronavirus disease 2019 (COVID-19), and hematological changes can persist after the acute COVID-19 phase. This study aimed to evaluate these hematological laboratory markers, linking them to clinical findings and long-term outcomes in patients with long COVID. This cross-sectional study selected participants from a ‘long COVID’ clinical care program in the Amazon region. Clinical data and baseline demographics were obtained, and blood samples were collected to quantify erythrogram-, leukogram-, and plateletgram-related markers. Long COVID was reported for up to 985 days. Patients hospitalized in the acute phase had higher mean red/white blood cell, platelet, and plateletcrit levels and red blood cell distribution width. Furthermore, hematimetric parameters were higher in shorter periods of long COVID than in longer periods. Patients with more than six concomitant long COVID symptoms had a higher white blood cell count, a shorter prothrombin time (PT), and increased PT activity. Our results indicate there may be a compensatory mechanism for erythrogram-related markers within 985 days of long COVID. Increased levels of leukogram-related markers and coagulation activity were observed in the worst long COVID groups, indicating an exacerbated response after the acute disturbance, which is uncertain and requires further investigation.

## 1. Introduction

The severe acute respiratory syndrome coronavirus 2 (SARS-CoV-2) is responsible for the coronavirus disease 2019 (COVID-19), which has millions of confirmed cases worldwide. SARS-CoV-2 infection can affect the respiratory tract and other organ systems, resulting in various multi-organ complications [1,2,3].

Although some patients recover from COVID-19 without complications, studies have shown that many patients experience various persistent impairments even after 4 weeks from the onset of acute symptoms. This condition is commonly referred to as ‘long COVID’, and it causes a variety of symptoms [4,5,6,7,8,9,10].

Individuals who survived previous coronavirus epidemics, such as the 2003 SARS epidemic and the 2012 Middle East Respiratory Syndrome (MERS) outbreak, exhibited persistent symptoms following the acute phase of infection, similar to those observed in long COVID. This observation heightens concerns over the potential long-term clinical consequences of COVID-19 [11,12]

Among the various clinical and laboratory manifestations of the disease described throughout the COVID-19 pandemic, investigations of the hematological profile have proven important in the risk assessment of severe cases of COVID-19. Patients with an unfavorable disease evolution have hypercoagulable states and changes in platelet, leucocyte, and erythrocyte counts, indicating a poor prognosis; these parameters should be observed carefully [6,13,14,15].

Some studies indicate that the hematological profile may remain altered during long COVID, with thrombotic events and leukocyte count imbalances observed [13,14,15]. Furthermore, while long COVID symptoms vary, abnormalities in hematological markers, particularly clotting parameters, related to COVID-19 infection are believed to play a role in the pathogenesis of long COVID symptoms [16]. These findings highlight the importance of monitoring blood parameters in patients with long COVID and the need for continued research to better understand the mechanisms underlying these hematological changes.

Therefore, this study aimed to evaluate hematological laboratory markers, linking them to clinical findings and long-term outcomes in patients with long COVID. The study seeks to explore how this relationship occurs in long COVID patients, particularly in scenarios lasting over a year. This is crucial in the Amazon region, where literature on this topic is limited. Describing the prevalence of abnormalities in erythrogram, leukogram, and plateletgram markers could help identify the hematological profile of long COVID patients and indicate those at future risk.

Our results suggest a potential compensatory mechanism for erythrogram-related markers within 985 days of long COVID. Increased leukogram-related markers and increased coagulation activity were observed in the worst long COVID groups.

## 2. Materials and Methods

This observational, cross-sectional, and prospective study was approved by the Ethics Committee for Research Involving Human Beings of the State University of Pará (opinion 4.252.664). This study followed the principles of the Declaration of Helsinki and the Strengthening the Reporting of Observational Studies in Epidemiology guidelines supporting the construction of this article. Written consent was obtained from all participants.

Adults (aged ≥18 years) of both sexes enrolled in a clinical care program for patients with long COVID, conducted in the Amazon region of Brazil between March 2020 and June 2022, following an open call. A total of 260 patients voluntarily enrolled in the program. The diagnosis of long COVID was carried out using the following criteria: (a) confirmation of acute symptomatic infection by SARS-CoV-2 by reverse transcriptase real-time polymerase chain reaction amplification—acute symptoms needed to be consistent with COVID-19 and not attributable to any other cause and (b) presence of at least one prolonged symptom of COVID-19 (post-acute COVID-19), such as fatigue, dyspnea, cough, chest pain, muscle pain or weakness, headache, insomnia, visual disturbances, tremor, loss of balance, lower limb edema, arthralgia, palate, and/or olfactory disorders, that cannot be assigned to another cause, for at least four weeks (28 days) after the acute onset of symptoms.

None of the included patients had clinically relevant infectious conditions that could confuse the interpretation of the variables studied here. In vaccinated patients, the time interval from the last vaccination to the collection of blood samples ranged from 49 to 377 days. Thus, the 260 patients were divided into groups based on the following criteria: ‘hospitalization in the acute phase’; ‘long COVID period’; and ‘number of long COVID symptoms’, which allowed comparisons, correlations, and associations to be carried out in the studied population (Figure 1).

The first stage of the collection consisted of venipuncture for blood collection, with 3 mL of blood collected in two Vacuette^®^ tubes each (Greiner Bio-One, Kremsmünster, Austria) after patients fasted for at least 8 h. The blood was collected into the following tubes: (a) tubes containing ethylenediaminetetraacetic acid (EDTA) anticoagulant for whole-blood analysis, including quantification of red blood cells (RBCs), hemoglobin, hematocrit, mean corpuscular volume (MCV), mean corpuscular hemoglobin (MCH), mean corpuscular hemoglobin concentration (MCHC), red blood cells distribution width (RDW), erythrocyte sedimentation rate (ESR), white blood cells (WBCs), neutrophils, eosinophils, basophils, monocytes, lymphocytes, platelets, plateletcrit, mean platelet volume, and platelet distribution width (PDW); and (b) tubes containing sodium citrate for coagulation analysis, including prothrombin time (PT), PT activity, and activated partial thromboplastin time (aPTT). The respective reagent manufacturers adopted the standard reference ranges for the examinations mentioned above (Appendix A). The second collection stage, which occurred within a maximum of 24 h after the first stage, consisted of face-to-face interviews to obtain clinical and baseline demographic data of the participants, such as sex, age, presence of comorbidities or concomitant infections, date of acute onset of symptoms, self-reported long COVID symptomatology, medications used, hospital admission in the COVID-19 acute phase, and length of stay.

Whole blood samples in tubes with EDTA were analyzed using the Zybio Z3 hematological analyzer (Zybio Inc., Chongqing, China), with the reagent line for hematological tests from Labtest (Lagoa Santa, Brazil). After 1 h of rest, the ESR was determined in whole blood samples. The blood samples in sodium citrate tubes were centrifuged at 3000 rpm for 5 min (Daiki 80-2B centrifuge, Ionlab, Araucária, Brazil), followed by the determination of clotting parameters in the HumaClot Junior coagulation analyzer, with its reagent kit line (HUMAN, Wiesbaden, Germany) for hemostasis tests.

GraphPad Prism™ software version 8.4.3 (GraphPad Software, San Diego, CA, USA) was used for statistical analysis. The D’Agostino–Pearson test was used to assess data normality, with mean and standard deviation (SD) used as dispersion measures to describe continuous variables. An analysis of variance (ANOVA) was used to compare variables with a normal distribution, and the Mann–Whitney U test was performed to compare variables with a non-normal distribution. Multiple logistic regression analysis was used to assess the associations between risk variables and long COVID outcomes. Furthermore, Pearson’s coefficient was used to evaluate the correlations between hospitalization duration and lymphocyte, monocyte, and neutrophil counts in the 89 patients with long COVID who were hospitalized during the acute COVID-19 phase. Statistical significance was defined as a two-tailed *p*-value of <0.05.

## 3. Results

Of the 260 patients selected, the majority were female (n = 166) and not elderly (n = 198), with 34.2% being hospitalized for acute COVID-19 (n = 89). Fatigue (n = 181), dyspnea (n = 176), and muscle weakness (n = 159) were the most commonly self-reported long COVID symptoms. The mean long COVID period was 308.1 days (SD, 171.5), with a mean of approximately six concomitant symptoms (mean ± SD, 6.1 ± 3.3). The longest reported long COVID period was 985 days. Most of the mean laboratory levels were within the reference ranges; however, the mean ESR levels increased, and the mean PDW levels were below the minimum adopted threshold (Table 1 and Appendix A). Only four patients (1.5%) presented with thrombocytopenia (platelet count, <150,000/mm^3^).

Hospitalized patients in the acute COVID-19 phase presented with increased RBCs, WBCs, eosinophils, lymphocytes, and platelets and increased percentages of RDW and plateletcrit than non-hospitalized patients (n = 171). In patients with up to 90 days of long COVID (n = 31), the MCV, MCH, and MCHC levels were lower than those in patients with long COVID for >90 days (n = 229), which can also be observed when comparing patients with up to 180 days (n = 60) and those with >180 days of long COVID (n = 200). However, patients with up to 365 days of long COVID (n = 172) presented with increased MCV, MCH, and MCHC. The RBC count and RDW percentages were increased in patients with up to 90 and 180 days of long COVID. Patients with more than six concomitant symptoms (n = 119) presented with increased levels of WBCs, neutrophils, and lymphocytes, shorter PTs, and higher PT activity than patients with less than six concomitant symptoms (Table 2).

Shorter periods of long COVID, such as up to 90 or 365 days, were associated with the female sex, acute hospitalization, and an increased platelet count. However, having long COVID for >1 year was associated with a higher lymphocyte count and low hemoglobin levels. The female sex and acute hospitalization were associated with more than six concomitant symptoms (Table 3). In hospitalized patients (n = 89), the neutrophil, monocyte, and lymphocyte levels were correlated with the hospitalization period (Figure 2).

## 4. Discussion

Our main findings suggest that the study population had, on average, a positive recovery in the hematological profile, even if symptoms of long COVID were reported for up to 985 days. Among the 89 hospitalized patients, there were higher mean levels of red/white blood cells, platelets, and percentages of plateletcrit and RDW. In addition, hematimetric parameters, such as MCV, MCH, and MCHC, were higher in shorter periods of long COVID than in longer periods. Patients with more than six concomitant long COVID symptoms had a higher WBC count, lower PT, and increased PT activity than patients with less than six concomitant long COVID symptoms. More than 1 year of long COVID was associated with low hemoglobin levels and increased lymphocyte counts.

Evidence indicates that thrombocytopenia and anemia influence the severity of acute COVID-19 [17,18,19], increasing the risk of fatal outcomes. Furthermore, even if limited to one week, immune thrombocytopenia was reported 4 weeks after the onset of COVID-19 symptoms [13], while anemia has been described as one of the possible unfavorable outcomes of long COVID up to 6 months after the acute phase [6]. However, in this study, hospitalized patients during the acute phase of COVID-19, which indicates a more severe acute involvement, had higher levels of hematological markers, such as RBC count, platelet count, and plateletcrit percentage, in the long COVID phase than those who were not hospitalized. This suggests homeostatic compensation in the convalescence phase after hospital discharge, which is further corroborated by the hematimetric parameters, such as MCV, MCH, and MCHC, which tend to be higher in longer periods of long COVID. This is observed at least until the first year of long COVID, similar to the higher RBC count and RDW percentages in lower periods of long COVID.

Abnormalities in the WBC count during the acute phase of COVID-19 indicate the severity of the illness and aid in risk assessment for the patient [20,21]. Specifically, low levels of lymphocytes may predict a severe or fatal outcome because they indicate an inadequate immune response to the virus. However, normalization of the lymphocyte levels suggests the patient is in the recovery phase. Furthermore, elevated counts of neutrophils and monocytes have been associated with severe cases of COVID-19 due to their relationship with hyperinflammation. They may indicate poor clinical outcomes, such as requiring intensive care unit admission. Leukocytosis has been linked to a more severe acute scenario [22,23,24,25,26,27,28].

Herein, in both the worse outcome groups—patients with more than six simultaneous long COVID symptoms and patients hospitalized in the acute phase—an increase in WBC count was observed, with higher lymphocyte levels and increased coagulation activity when compared with their counter groups. Furthermore, in 89 patients previously hospitalized for acute COVID-19, there was a correlation between their length of hospitalization and lymphocyte, monocyte, and neutrophil levels, with longer hospital stays correlating to higher levels of these markers. Considering the behavior of the erythrogram in this study, these results suggest a response that exacerbates the effects of the acute disturbance emerging in the long COVID phase [14]; however, further investigations are needed.

This study had several limitations. Other study groups, such as the control group of patients without long-term symptoms, would allow interesting comparisons regarding the hematological status of patients with long COVID. Furthermore, assessing the patient’s hematological profile at the time of the COVID-19 acute phase could indicate the course of the hematological implications of long COVID, which was not performed in this study. However, to our knowledge, this is the first study to provide clues about hematological laboratory markers of patients with long COVID up to 985 days while relating abnormalities in these markers with clinical outcomes. Additionally, the lack of similar investigations in the Amazon region emphasizes the significance of this research.

Therefore, observing abnormalities in hematologic screening markers can provide several benefits for understanding long COVID. This approach can act as a starting point for identifying severe hematologic diseases long after an acute SARS-CoV-2 infection. It can also serve as a basis for future longitudinal studies that assess changes in hematological profiles over time, and as an initial observation of the prevalence of laboratory abnormalities in patients with worse clinical scenarios. These findings can indicate potential risk patterns involved in the post-acute phase and assist in risk stratification. Consequently, this study can help clinicians identify patients with a more severe presentation of long COVID, enabling better management of this condition.

## 5. Conclusions

We evaluated how the laboratory hematological levels behave in patients presenting with long-term COVID-19 symptoms for up to 985 days after the acute COVID-19 phase. Hospitalized patients presented with increased levels of hematological parameters, such as higher mean levels of RBCs and platelets, suggesting a benign compensation after the acute COVID-19 phase. Increased WBCs, lymphocytes, and a shorter PT were observed in the groups with worse long COVID outcomes, such as the hospitalization group or the group with more simultaneous symptoms, indicating an exacerbated response to acute involvement; however, this is not certain.

These findings provide an interesting insight into the hematological profile of long COVID while promoting reflection for future studies. It is suggested that future investigations should address the issue of increased leucocyte and lymphocyte levels and increased coagulation activity, even 4 weeks after the onset of symptoms. In addition, follow-up studies of long COVID are important, as they would help to better visualize long-term hematological changes.

## Figures and Tables

**Figure 1 viruses-15-00879-f001:**
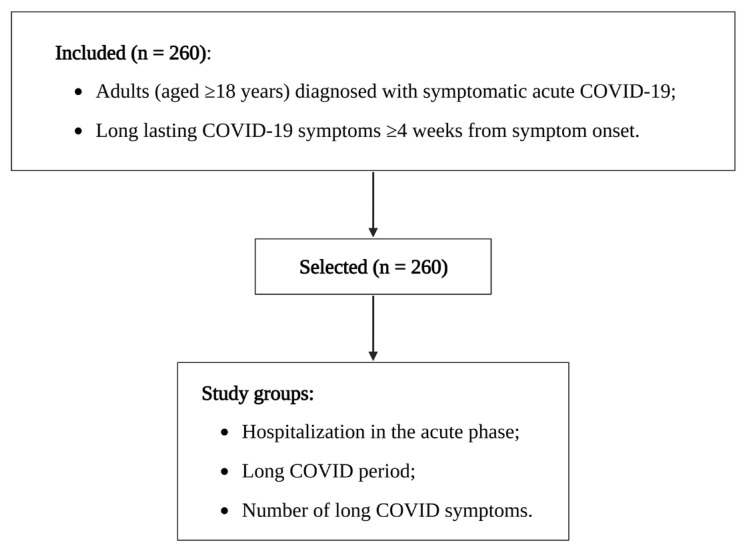
Study flowchart for recruitment and group allocation. Created with BioRender.com (www.biorender.com. accessed on 20 February 2023).

**Figure 2 viruses-15-00879-f002:**
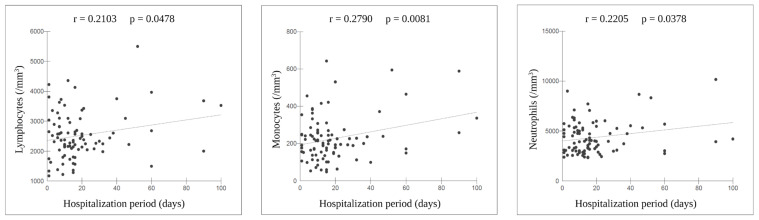
Correlation between the lymphocytes, monocytes, and neutrophils levels and hospitalization period among the 89 hospitalized patients. r, Pearson’s correlation coefficient. Created with BioRender.com (www.biorender.com. accessed on 20 February 2023).

**Table 1 viruses-15-00879-t001:** Clinical and laboratory profile of the study population.

Variable	n = 260
**Female, n (%)**	166 (63.8)
Age, mean ± SD, years	49.2 ± 12.7
≥60 years old, n (%)	62 (23.8)
Hospitalized in the acute phase ^(a)^, n (%)	89 (34.2)
Hospitalization period ^(b)^, mean ± SD, days	18.8 ± 19.4
Fatigue ^(c)^, n (%)	181 (69.6)
Dyspnea ^(c)^, n (%)	176 (67.6)
Muscle weakness ^(c)^, n (%)	159 (61.1)
Muscle pain ^(c)^, n (%)	149 (57.3)
Headache ^(c)^, n (%)	136 (52.3)
Loss of balance ^(c)^, n (%)	119 (45.7)
Insomnia ^(c)^, n (%)	113 (43.4)
Chest pain ^(c)^, n (%)	110 (42.3)
Visual disturbances ^(c)^, n (%)	105 (40.3)
Cough ^(c)^, n (%)	91 (35)
Tremor ^(c)^, n (%)	79 (30.3)
Lower limb edema ^(c)^, n (%)	73 (28)
Palate and olfactory disorders ^(c)^, n (%)	62 (23.8)
Arthralgia ^(c)^, n (%)	41 (15.7)
Number of long COVID symptoms, mean ± SD	6.1 ± 3.3
≤6 symptoms, n (%)	141 (54.2)
>6 symptoms, n (%)	119 (45.7)
Long COVID period, mean ± SD, days	308.1 ± 171.5
≤90 days, n (%)	31 (11.9)
≤180 days, n (%)	60 (23)
>365 days, n (%)	88 (33.8)
RBCs, mean ± SD, millions/mm³	4.7 ± 0.6
Hemoglobin, mean ± SD, g/dL	13.1 ± 1.3
Hematocrit, mean ± SD, %	39.2 ± 4.1
MCV, mean ± SD, fL	84 ± 9
MCH, mean ± SD, pg	28.3 ± 3.3
MCHC, mean ± SD, %	33.6 ± 1.2
RDW, mean ± SD, %	13.7 ± 1.3
ESR, mean ± SD, mm	40.4 ± 26.4
WBCs, mean ± SD, thousands/mm³	6.8 ± 2.1
Neutrophils, mean ± SD, thousands/mm³	4.1 ± 1.4
Eosinophils, mean ± SD, /mm³	220.4 ± 119.3
Basophils, mean ± SD, /mm³	30.6 ± 37
Monocytes, mean ± SD, /mm³	209.4 ± 111.1
Lymphocytes, mean ± SD, thousands/mm³	2.3 ± 1
Platelets, mean ± SD, thousands/mm³	302.7 ± 87.8
MPV, mean ± SD, fL	8.5 ± 1
Plateletcrit, mean ± SD, %	0.256 ± 0.073
PDW, mean ± SD, %	16 ± 6.4
PT, mean ± SD, s	12.3 ± 1.1
PT activity, mean ± SD, %	96.1 ± 23
aPTT, mean ± SD, s	30.2 ± 5.7
**Total**, n (%)	260 (100)

Values are expressed as the mean ± standard deviation (SD). (a) Heart, kidney, liver, or neurological diseases; (b) n = 89; (c) shown for ≥4 weeks from symptom onset. RBCs, red blood cells; MCV, mean corpuscular volume; MCH, mean corpuscular hemoglobin; MCHC, mean corpuscular hemoglobin concentration; RDW, red cell distribution width; ESR, erythrocyte sedimentation rate; WBCs, white blood cells; MPV, mean platelet volume; PDW, platelet distribution width; PT, prothrombin time; aPTT, activated partial thromboplastin time.

**Table 2 viruses-15-00879-t002:** Comparison of the hematological levels and clotting parameters in the long COVID outcome groups.

Variable	Hospitalized in the Acute Phase			Long COVID Period									Number of Long COVID Symptoms		
	Yes	No	*p* *	≤90 Days	>90 Days	*p* *	≤180 Days	>180 Days	*p* *	≤365 Days	>365 Days	*p* *	≤6	>6	*p* *
RBCs, mean ± SD, millions/mm³	4.8 ± 0.6	4.6 ± 0.6	0.0179	4.9 ± 0.5	4.6 ± 0.6	0.0104	4.8 ± 0.6	4.6 ± 0.6	0.0175	4.6 ± 0.6	4.7 ± 0.6	0.0943	4.6 ± 0.6	4.7 ± 0.6	0.7387
Hemoglobin, mean ± SD, g/dL	13.3 ± 1.2	13.1 ± 1.3	0.1644	13.3 ± 1.5	13.1 ± 1.3	0.6262	13.2 ± 1.3	13.1 ± 1.3	0.4144	13.2 ± 1.2	13 ± 1.4	0.2598	13.2 ± 1.4	13.1 ± 1.2	0.5828
Hematocrit, mean ± SD, %	39.5 ± 4	39 ± 4.1	0.6307	40.2 ± 3.6	39.1 ± 4.2	0.1340	39.8 ± 4.1	39 ± 4.1	0.2023	39.3 ± 4.1	39.1 ± 4.1	0.7379	39.3 ± 4.3	39 ± 3.8	0.5742
MCV, mean ± SD, fL	83.3 ± 8.5	84.3 ± 9.2	0.0502	81.8 ± 5.7	84.3 ± 9.3	0.0160	82.6 ± 6.5	84.4 ± 9.5	0.0317	85.2 ± 7.5	81.5 ± 10.9	0.0018	84.9 ± 8.3	82.8 ± 9.5	0.0849
MCH, mean ± SD, pg	28.1 ± 3.8	28.4 ± 3.1	0.0601	27.2 ± 2.2	28.5 ± 3.4	0.0053	27.6 ± 2.8	28.5 ± 3.5	0.0123	28.8 ± 3.2	27.4 ± 3.4	0.0019	28.6 ± 3.4	28 ± 3.2	0.0548
MCHC, mean ± SD, %	33.6 ± 1.3	33.5 ± 1.2	0.4145	33 ± 0.8	33.6 ± 1.2	0.0023	33.3 ± 1.1	33.6 ± 1.2	0.0150	33.7 ± 1.3	33.3 ± 1.1	0.0270	33.6 ± 1.3	33.5 ± 1.1	0.5651
RDW, mean ± SD, %	14.1 ± 1.1	13.4 ± 1.4	<0.0001	14.2 ± 0.8	13.6 ± 1.4	<0.0001	14.1 ± 1	13.5 ± 1.4	<0.0001	13.6 ± 1.4	13.7 ± 1.2	0.8452	13.7 ± 1.6	13.6 ± 1	0.2541
ESR, mean ± SD, mm	41.6 ± 28.4	39.8 ± 25.3	0.8307	44.9 ± 22.9	39.8 ± 26.8	0.1164	44.9 ± 28.8	39 ± 25.5	0.1614	40 ± 24.8	41.2 ± 29.3	0.8974	41.4 ± 27.5	39.2 ± 25	0.6760
WBCs, mean ± SD, thousands/mm³	7.2 ± 2.2	6.6 ± 2.1	0.0179	6.6 ± 1.6	6.9 ± 2.2	0.7055	6.9 ± 2.2	6.8 ± 2.1	0.8166	6.8 ± 1.9	6.9 ± 2.5	0.7137	6.5 ± 2.1	7.2 ± 2.2	0.0083
Neutrophils, mean ± SD, thousands/mm³	4.3 ± 1.6	4 ± 1.4	0.2369	3.9 ± 1.1	4.1 ± 1.5	0.6755	4.2 ± 1.5	4.1 ± 1.4	0.6646	4.1 ± 1.3	4 ± 1.7	0.3108	3.9 ± 1.3	4.3 ± 1.6	0.0356
Eosinophils, mean ± SD, /mm³	232.7 ± 105.7	214 ± 125.7	0.0366	261.4 ± 85.6	214.9 ± 122.3	0.0017	232.6 ± 108.2	216.8 ± 122.5	0.1303	228.3 ± 124.1	205.1 ± 108.6	0.1653	216.5 ± 116.7	225.1 ± 122.8	0.6985
Basophils, mean ± SD, /mm³	28.4 ± 34.4	31.8 ± 38.4	0.4869	37 ± 36.3	29.7 ± 37.1	0.2224	29.3 ± 33.6	31 ± 38.1	0.9104	33.6 ± 36.3	24.8 ± 38	0.0268	29.3 ± 37	32.1 ± 37.2	0.4484
Monocytes, mean ± SD, /mm³	225.7 ± 121.3	200.9 ± 104.7	0.0938	164.4± 76.3	215.5 ± 113.7	0.0130	213.8 ± 142.8	208.1 ± 100	0.5884	204.6 ± 116.2	218.7 ± 100.3	0.0729	193.7 ± 89.2	228 ± 130.3	0.0703
Lymphocytes, mean ± SD, thousands/mm³	2.5 ± 0.7	2.2 ± 1	<0.0001	2.2 ± 0.6	2.3 ± 1	0.4582	2.2 ± 0.8	2.3 ± 1	0.6933	2.2 ± 0.8	2.4 ± 1.2	0.1387	2.2 ± 1.1	2.4 ± 0.8	0.0281
Platelets, mean ± SD, thousands/mm³	320 ± 101.1	293.6 ± 78.8	0.0486	339 ± 98.8	297.7 ± 85.2	0.0225	313.3 ± 98.5	299.4 ± 84.3	0.4683	295.7 ± 82.4	316.2 ± 96.4	0.1260	297.6 ± 87.4	308.6 ± 88.2	0.1854
MPV, mean ± SD, fL	8.5 ± 0.8	8.6 ± 1.1	0.7327	8.5 ± 0.9	8.5 ± 1	0.6287	8.5 ± 0.8	8.5 ± 1.1	0.7758	8.5 ± 1	8.5 ± 1	0.6898	8.6 ± 1	8.5 ± 1	0.3808
Plateletcrit, mean ± SD, %	0.270 ± 0.079	0.249 ± 0.068	0.0244	0.290 ± 0.091	0.251 ± 0.069	0.0151	0.267 ± 0.086	0.252 ± 0.068	0.3336	0.251 ± 0.073	0.266 ± 0.071	0.1437	0.254 ± 0.072	0.259 ± 0.073	0.4371
PDW, mean ± SD, %	16.1 ± 0.7	15.9 ± 7.9	0.0819	16.1 ± 0.3	16 ± 6.8	0.3911	15.8 ± 1.3	16 ± 7.3	0.4211	15.4 ± 1.8	17 ± 10.7	0.8994	16.4 ± 8.6	15.5 ± 1.6	0.1396
PT, mean ± SD, s	12.3 ± 1	12.4 ± 1.2	0.2488	12.6 ± 1.1	12.3 ± 1.1	0.1864	12.4 ± 1.1	12.3 ± 1.1	0.7004	12.3 ± 1.2	12.4 ± 0.9	0.7998	12.4 ± 1.1	12.2 ± 1.1	0.0380
PT activity, mean ± SD, %	96.3 ± 19.2	95.9 ± 24.7	0.2439	91.5 ± 18.5	96.7 ± 23.5	0.3222	95.1 ± 19.5	96.4 ± 23.9	0.8425	97.2 ± 24.7	93.9 ± 19	0.7544	94.6 ± 23.9	97.8 ± 21.8	0.0443
aPTT, mean ± SD, s	29.5 ± 4.9	30.6 ± 6.1	0.1334	29.6 ± 6.4	30.3 ± 5.6	0.5338	29.6 ± 5.4	30.4 ± 5.8	0.3875	30.3 ± 5.4	30.1 ± 6.2	0.9229	30.3 ± 4.5	30.1 ± 6.9	0.4766
Total, n (%)	89 (34.2)	171 (65.7)	−	31 (11.9)	229 (88)	−	60 (23)	200 (76)	−	172 (66.1)	88 (33.8)	−	141 (54.2)	119 (45.7)	−

Values are expressed as the mean ± standard deviation (SD); * *p*-value, ANOVA and Mann–Whitney U tests were used to compare normal and non-normal continuous variables, respectively. RBCs, red blood cells; MCV, mean corpuscular volume; MCH, mean corpuscular hemoglobin; MCHC, mean corpuscular hemoglobin concentration; RDW, red cell distribution width; ESR, erythrocyte sedimentation rate; WBCs, white blood cells; MPV, mean platelet volume; PDW, platelet distribution width; PT, prothrombin time; aPTT, activated partial thromboplastin time.

**Table 3 viruses-15-00879-t003:** Association between the hematological and clotting abnormalities and long COVID outcomes.

Variables	Long COVID Outcomes								
	Long COVID Period >90 Days (n = 229)			Long COVID Period >365 Days (n = 88)			Number of Long COVID Symptoms >6 (n = 119)		
	Coefficient	*p*-Value	Odds Ratio	Coefficient	*p*-Value	Odds Ratio	Coefficient	*p*-Value	Odds Ratio
Female gender	0.9822	0.0337	2.6703	−0.4808	0.1180	0.6183	0.7387	0.0179	2.0931
Age ≥ 60 years	−0.3446	0.4360	0.7085	−0.3289	0.3320	0.7197	−0.6762	0.0384	0.5086
Hospitalization in acute phase	−0.9319	0.0395	0.3938	−0.6692	0.0399	0.5121	1.2739	<0.0001	3.5748
Long COVID period, ≤90 days	−	−	−	−	−	−	−0.1852	0.6724	0.8310
RBCs < 4 million/mm^3^	1.8704	0.0906	6.4909	−0.7894	0.1112	0.4541	−0.1674	0.6939	0.8459
Hemoglobin < 12 g/dL	−0.2671	0.6735	0.7656	1.0673	0.0057	2.9076	−0.2053	0.5918	0.8144
Neutrophils > 5 thousands/mm^3^	−0.0267	0.9585	0.9737	−0.6306	0.0718	0.5322	0.4955	0.1261	1.6414
Lymphocytes > 2.5 thousands/mm^3^	1.1104	0.0508	3.0355	0.6254	0.0495	1.8690	0.0559	0.8567	1.0575
Platelets > 450 thousands/mm^3^	−2.1664	0.0012	0.1146	0.3219	0.5889	1.3797	−0.3558	0.5662	0.7006

Probability prediction of long COVID clinical and laboratory outcomes using multiple logistic regression. RBCs, red blood cells.

## Data Availability

The data supporting the study’s findings are available on request from the corresponding author, L.F.M.F. The data are not publicly available because they contain information that could compromise the privacy of the participants.

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
