# Peer review of "Evaluation of the Hematological Patterns from Up to 985 Days of Long COVID: A Cross-Sectional Study"

_viruses, 2023, doi:10.3390/v15040879_

Round 1

Reviewer 1 Report

This research can be accepted for publication in the present form

Reviewer 2 Report

Thank you for the opportunity to review the manuscript “Evaluation of the haematological patterns from up to 985 days of long COVID: a cross-sectional study”  (viruses-2274879).

Clinical data and baseline demographics were obtained, and blood samples were collected for quantification of erythrogram-, leukogram-, and plateletgram-related markers in the context of Long-Covid. Investigations into long covid are very important because of the extent of the disease.

However, some parts of the review must be carefully revised.

One of my concerns related to the current observational study continues to be related to the introduction literature. Greater details about previous studies results are needed than currently provided that builds the case for having conducted the current study. This could also strengthen the discussion, as it is quite common to refer to findings from those studies relative to the current study findings in the discussion and conclusions sections.

Authors should also refer to previous studies on antecedent diseases of Severe acute respiratory syndrome (SARS).

There is no research question, no hypothesis at the end of introduction.

Discussion:  It would also be helpful to explain how the current study differs from previous studies in more detail. This section should be streamlined to focus specifically on key findings.

Authors should include a table with the most important results of their observational study.

What is the implication for further studies?

What contribution does this study make to medical progress in combating long covid?

Authors should provide several concrete examples.

Reviewer 3 Report

1: Authors should add more references.

2: Authors should perform minor spell and grammar check

3: Figure 2 should be more clear.

4: Keywords should be at least 5.

5: Introduction should be expanded.

6: Results should be discussed in more detail.
